**Title: Monitoring researchers profiles on Wikidata: a comparative analysis of country-specific metrics**

**Author(s):** Alessandro Marchetti

## Author biography

Note: The original guidelines on the Wiki page suggested 300 words, not 300 characters. I am keeping the previous version as is.

## Long version

*Alessandro Marchetti (born 1983) is a Wikimedian with a scientific academic and professional background.*

*He pursued his studies at the University of Pisa and Scuola Normale Superiore, further enhancing his expertise at the École Normale Supérieure de Lyon. He was a laboratory technician in the field of oceanography and continued with a postdoctoral research position at Zhejiang University, where he contributed to the field of soft matter. He is now technologist (Project manager) at the Department of Biology of the University of Pisa, where he is active in the management and the D.E.C.O. activities of E.U.-funded projects.*

*Beyond his scientific career, he is involved in the Wikimedia community. He is a member of the WikiClassics User Group, Wikimedia Italia, and Wikimedia Switzerland. His past affiliations include active participation in the WikiDonne user group and Wikimedia Portugal. He is the coordinator for Wiki Loves Monuments in Tuscany and the international contest Wiki Science Competition.*

*Currently, Alessandro is primarily active with Wikimedia Switzerland, where he continues to promote and support the dissemination of free knowledge, his main field is currently the use of Wikidata for bibliometry.*

## Short version

*Alessandro Marchetti (b. 1983) works at the University of Pisa's Department of Biology. With a background in oceanography and soft matter, he manages EU-funded projects. Active in Wikimedia Switzerland, his corrent focus is Wikidata and bibliometry. He coordinates Wiki Loves Monuments in Tuscany.*

## Author(s) bibliography (Wiki-related):

- https://www.wikidata.org/wiki/Wikidata:WikiCite/Researchers_in_Switzerland
- https://www.wikidata.org/wiki/Wikidata:Sistema_Cultura
- https://www.wikidata.org/wiki/Wikidata:Testi_latini
- Le immagini per la didattica e la divulgazione: ricerca on-line e copyright, CnS-La Chimica nella Scuola, January-March 2012, 61-63

**Title: Monitoring researchers profiles on Wikidata: a comparative cnalysis of country-specific metrics**

**Keywords:** open-access bibliometry, Wikicite, data quality, digital identity, SPARQL

**TL;DR:** This study analyzes the distribution, quality, and completeness of researcher metadata on Wikidata across countries

**Abstract**

*Wikidata, the open and structured data repository of Wikimedia projects, shows an increasingly significant potential as a platform for organizing and sharing metadata in the academic ecosystem. One aspect is the information related to researchers and authors of peer-reviewed content.*

*This analysis details the use of the QLever SPARQL query service to describe and monitor researchers' items on Wikidata, and related fundings. By analyzing the distribution and quality of key metrics such as statements and external identifiers, the study provides a comparative analysis across profiles linked to different countries, focusing on researchers' educational backgrounds, affiliations, and employers.*

*Key variations in data coverage, completeness, and quality across different academic ecosystems are investigated, specifically:*

- *Item distribution: how many researcher items can be associated with each country?*

- *Statement distribution: which properties are most added to researcher items?*

- *Identifier distribution: which databases are commonly connected to Wikidata?*

*This comparative approach enables us to reveal patterns and potential biases in how research communities are represented on Wikidata, shedding light on the global academic landscape.*

*These findings have potential implications for data accuracy, reliability, and equity on Wikidata, as well as for broader discussions on the digital identity of researchers. We propose strategies for further improving data quality, enhancing representation, and fostering collaboration between all stakeholders, primarily Wikidata editors, academic institutions, and research organizations based on existing active projects. By addressing these challenges, we aim to strengthen the framework for supporting the academic community on Wikidata and contribute to building a more comprehensive and accurate global knowledge base.*

