# OpenReview forum: "Monitoring researchers profiles on Wikidata: a comparative analysis of country-specific metrics"
_wikimedia.it/Wikidata_and_Research/2025/Conference — Submitted to WD&R_

### Official Review · ~Federico_Morando1 · 2025-01-19
**Relevant topic, competent author**

**Originality:** 5
**Impact:** 5
**Confidence:** 4

**Review:**

The proposed paper analyzes the distribution, quality, and completeness of metadata related to researchers and authors of peer-reviewed content on Wikidata across different countries.

The topic is undoubtedly relevant, and the author appears to be well-versed in the subject matter. I also find the approach to be original, as it places a greater emphasis on researchers rather than research outputs.

Based on the content of the abstract and the attached PDF, it is challenging to fully evaluate the scientific methodology and, consequently, the overall scientific quality. However, the submission appears to be on par with, if not superior to, other submissions. Therefore, I recommend its acceptance.

**Compliance:**

5

**Scientific Quality:**

4

---

### Official Review · ~Iolanda_Pensa1 · 2025-01-21
**A topic which directly links Wikidata and research**

**Originality:** 4
**Impact:** 5
**Confidence:** 4

**Review:**

The proposal looks at data related to researchers on Wikidata. It is a topic directly related to "Wikidata and research" (the theme of this conference), its relevance is easily understandable by researchers and academia and, somehow, it provides an overview of the status of the research ecosystem on Wikimedia. It is also connected to many existing repositories (bibliographic, orcid, research projects by national funds and EU...) and it can stress the importance of interoperability, the reuse of data and more open licenses and tools used in those repositories. It can also be a presentation connected to the topic of research assessment and interesting to facilitate discussions about the potential roles of Wikidata in research assessment and the publication of relevant data for research assessment on Wikidata.
Looking forward to listen to it :)

**Compliance:**

5

**Notes:**

It would be fantastic if Alessandro Marchetti could also highlight how uploads have been made in different countries if the information is available. It would be extremely important to understand If specific projects - like the ones supported by Wikimedia CH or the project implemented in Italy - are determinants (and also needed in other countries) or if uploads connected to ORIC or research projects (and their funders) have or can have an important impact. This would allow us to better understand which strategies could be replicated in different countries to increase and improve data. I would be personally very interested also in the presence of images (many universities make photographic campaigns of their collaborators and it is really a pity that those are not available in CC0 or open licenses) and the links with research projects.

**Scientific Quality:**

4

---

### Decision · Program_Chairs · 2025-02-05

**Decision:**

Withdrawn

**Comment:**

ritirato